

# In situ observations of infragravity wave directionality at nearshore coastal sites

Takehiko Nose[1], Alexander Babanin[2], and Kevin Ewans[3]

[1]Faculty of Science, Engineering, and Technology, Swinburne University of Technology, Hawthorn, Victoria 3122 Australia
[2]Department of Infrastructure Engineering | Melbourne School of Engineering, The University of Melbourne, Victoria 3010 Australia
[3]MetOcean Research Ltd, New Plymouth, 4310 New Zealand

*Correspondence to:* Takehiko Nose (tnose@swin.edu.au)

**Abstract.** 'Infragravity waves' is a term used to collectively describe surface gravity waves with periods arbitrarily between 30 and 300 s. In situ observations of infragravity waves at nearshore sites are scarce, and the directionality of the wave field has not received much attention in the past. This paper details a systematic directional analysis of experimental infragravity wave data. Through applying conventional and new directional analysis methods, qualitative and some quantitative characteristics

of infragravity wave directions have been resolved. The analysis has found that infragravity waves have a bimodal directional structure with the dominant energy distributed in the propagation sector incident to the coast. It has also been demonstrated that mean infragravity wave directions can be derived, and there is evidence that the directional spreading of infragravity waves is correlated to their wind-generated wave counterparts. Using a numerical model, the qualitative findings were verified; however, contrary to the observations, the dominant direction of the modelled infragravity waves are in the propagation sector outward

from the coast. The results provide improved insights into the directionality of infragravity waves, but the disparity between the dominant directions in the model and observations remains to be resolved.

*Copyright statement.* TEXT

## 1 Introduction

Low frequency surface motions in the ocean are known as infragravity waves. Despite their name, infragravity waves are
arbitrarily defined as a group of surface gravity waves with periods greater than those of wind-generated waves, roughly between 30 and 300 s. The primary distinction between wind-generated waves and infragravity waves is that the wind-generated waves are forced directly by wind and atmospheric pressure, while infragravity wave energy is generated through the nonlinear energy transfers from wind-generated waves. Infragravity waves are broadly categorised into two types based on their origin of generation as 'bound' or 'free' infragravity waves. Bound waves are understood to be generated through a well-known
phenomenon of nonlinear wave-wave interactions, in which two wind-generated wave frequencies force secondary waves to both sum and difference frequencies. They are non-dispersive, phase-locked and found to be in anti-phase with the incident



carrier wave group. Free waves refer to the low frequency waves that are dispersive. Their origin is complex, and the physics of their generation is not comprehensively understood. They are either a liberated form of bound waves or directly forced from nonlinear phenomenon related to the time-varying break-point of a wind-generated wave breaking process (Symonds et al., 1982). Section 3 of Baldock (2012) provides a critical review of these theories relating to the evolution of free waves

for further reference. Regardless of their origin or generation, they can propagate in both incident and reflected directions (i.e., toward and outward from the coast), and depending on the incoming wave obliqueness and a geological setting (e.g., continental shelf width and bottom slope), they can escape to deep water as leaky waves or are refractively trapped within the continental shelf as edge waves (Herbers et al., 1995b).

Infragravity wave directional properties outside of the surf zone are seldom studied. This is partly because infragravity

wave signals are difficult to accurately measure due to the low steepness, having very long wavelengths and small amplitudes. Further, vastly different temporal and spatial scale between bound waves and free waves, as well as wind-generated waves and infragravity waves, cause inherent difficulties in obtaining reliable directional measurements. Directionality is one of the fundamental properties we study of ocean waves, but for infragravity waves, it is particularly interesting, as the waves of different modes or origin have varying directional characteristics, and in theory, they can be multi-directional.

To the author's knowledge, Herbers et al. (1995b) are the first to investigate the directional properties of low frequency waves outside of the surf zone. Using data collected from an array of pressure sensors in a nominal depth of 13 m at the U.S. Army Corps of Engineers, Coastal and Hydraulics Laboratory, Field Research Facility (FRF) location in Duck, North Carolina, they derived concurrent directional distributions for both wind-generated waves and infragravity waves. They observed that infragravity wave directional spreading is generally broader than the wind-generated wave counterparts and reflected energy

dominated the distributions. This is depicted in Fig. 6 of Herbers et al. (1995b). An obvious implication of a broad directional distribution is the presence of alongshore propagation of infragravity waves, which manifest in the form of edge waves that are refractively trapped within the continental shelf. Using the shallow water geometrical optics (WKB) theory, Herbers et al. (1995b) predicted that asymptotically far from shore on a mild monotonically alongshore uniform sloping seabed, free infragravity waves can be directionally isotropic owing to the presence of edge waves. Herbers et al. (1995a) further demonstrated

the multi-directional distributions of infragravity waves using the same data, as they estimated up coast and down coast energy fluxes for both waves. Figure 1 of Herbers et al. (1995a) illustrates these ratios, and it shows that infragravity wave propagation appears to be dependent on the wind-generated wave counterparts to some degree, e.g., up coast propagating wind-generated waves force predominantly up coast propagating reflected infragravity waves. However, the ratios of the up coast and down coast infragravity wave energy fluxes are more skewed toward unity than the equivalent wind-generated wave ratios. They in-

terpreted that the up coast propagating wind-generated waves force dominant up coast propagating infragravity waves but also some down coast propagating infragravity waves as well. Herbers et al. (1995b) also observed infragravity waves from remote sources in the absence of locally forced infragravity wave energy during benign sea states. Directional properties during these periods are characterised by bimodal directional distributions comprising narrower directional peaks than those locally forced, and the dominant energy was spread in the directional sector incident to the coast.





Reniers et al. (2010) conducted a study using experimental field data collected at the same FRF site during the year of 2005 utilising a linearised infragravity wave model called IDSB (Reniers et al., 2002). They evaluated directional spreading as part of the study and produced infragravity frequency integrated directional spectrograms for the month of April as shown in Fig. 3 of Reniers et al. (2010). In their plots, the broad and obscure nature of directional spreading is apparent. Although,

distinguishable energy distributions that are bimodal can be observed during the energetic sea states, as they have peaks in the shore-normal onshore and offshore propagation (corresponding to 0 and ±180 degrees). In contrast to the Herbers et al. (1995b) observations, Reniers et al. (2010) observational plot depicts that the infragravity wave energy is more prevalent in the propagation sector incident to the coast, but their numerical simulations produced more dominant energy distributed in the reflected wave sector. The observed dominant energy distributions in disparate directional sectors may be related to their

measurement depths; Herbers et al. (1995b) derived their directional spectra in the 13 m depth of water, and Reniers et al. (2010) obtained the directional data at a depth of 8 m.

Notwithstanding the apparent scarce nature of infragravity wave directional studies, these studies suggest that infragravity wave directional information can be obtained through conventional directional analysis methods. However, differing characteristics in the directional distributions at the same site suggest the derivation of infragravity wave directions is a complex

problem.

The ensuing Sect. 2 provides a description of the experimental infragravity wave data used in this study. Details of the systematic infragravity wave analysis are followed and reported in two parts; Sect. 3 examines an infragravity wave height correlation with the wind-generated wave integrated parameters, and Sect. 4 explores their directionality. Analysis results and discussions for both subjects are provided for each section to demonstrate the study findings of this experimental investigation,

and the paper concludes with a summary in Sect. 5.

## 2   Data description: site geography and instrumentation

Ideally, all sourced experimental data are collected in measurement campaigns dedicated to capturing infragravity wave signals with appropriate sampling configurations (e.g., longer sampling duration to collect sufficient infragravity wave samples). Experimental field data were sourced at four locations scattering across three ocean bodies as shown in Fig. 1. We sourced

five data sets with the total duration in excess of two years. Two dedicated infragravity wave data sets were provided by Shell Corporation. Beyond that however, such data sets are scarce compared to conventional wind-generated wave measurements and not readily available. Consequently, we broadened our search to include conventional wave data, typically those that are freely available in the public domain. While all data are described, the analysis discussions primarily focused on the FRF site and therefore constitute more details than the other sites. An overview of these data is given in Table 1.

As part of a previous joint industry project on infragravity waves, Shell Corporation obtained one year of infragravity wave data in 2005 at Duck, North Carolina from FRF, a well-known location/institution of experimental coastal data. The FRF site is an open straight coastline facing east northeast and exposed to a mix of swell propagating over the North Atlantic Ocean and locally generated wind waves. Approximate bed slope in the region is roughly 0.01 to a depth of around 10 m, followed by



more gentle gradients of around 0.0025 and 0.00075 to depths of around 16 and 26 m, respectively. The slopes were estimated based on the depths of the wave instruments. Figure 2 provides a schematic diagram of instruments deployed at the FRF site. To capture infragravity wave data, FRF deployed a coherent array of 15 bottom mounted pressure sensors located at a cross-shore distance of approximately 800 m from the shoreline in a nominal depth of around 8 m. The maximum distance between

the furthest co-located sensors within the array was around 250 m; its high spatial coverage and resolution makes this data set unique for obtaining directional properties in the infragravity frequency bands. Pressure data were collected in eight burst measurements per day, each containing 20,480 points at a 2 Hz sampling frequency, which is equivalent to around 170 minutes in duration. The array pressure data are supplemented with deeper water wind-generated wave data captured by a directional waverider buoy, WRB-16m, which measured XYZ displacements through double integration of acceleration recorded by an

on-board accelerometer. WRB-16m had a 1.28 Hz sampling frequency and collected data at a 2 hour and 50 minute burst on three hourly intervals. The WRB-16m wave gauge was located some 2.5 km offshore at a nominal depth of 16 m. Data files are segmented monthly; intermittent time series losses of the WRB-16m data, although rare, caused disorientation of time stamps within the data files. Consequently data files for the months of March, May, and July were omitted from the analysis.

Another set of data for this site is storm event based archival wave data; each event typically contains approximately two to

five days of measurements, and the data are freely shared in the public domain through the FRF archive website (U.S. Army Corps of Engineers, FRF). Nearshore wave measurement setup includes four Nortek AWACs at approximate depths of 5, 6, 8, and 11 m. In deeper water, directional waverider buoys were deployed at approximate depths of 16 and 24 m (refer to Fig. 2). The AWACs utilise a Doppler sonar technology to collect wave data at hourly intervals for around a 34 minute burst duration with sampling frequencies of 2 and 4 Hz for pressure/velocity and surface track data, respectively. The waverider buoys

sampled continuously at 1.28 Hz collecting XYZ displacements through double integration of recorded acceleration. These data constitute a diverse range of wave climate, including a number of hurricanes, with Event 19 containing data during the destructive Hurricane Sandy. Table 2 summarises the selected measurement periods and their respective instrument availability for this study.

Shell Corporation's data from previous projects were also obtained for the Baja site located on the south west coast of

California and the Lagos site in Nigeria. At the Baja site, two instruments at a nominal depth of 20 m were deployed in 2002; a Datawell GPS directional waverider buoy with a sampling frequency of 2 Hz collected infragravity waves, and wind-generated waves were captured by a bed mounted Teledyne RDI 300 kHz Workhorse ADCP, using a similar Doppler sonar technology to AWACs, configured to record wave data for approximately 34 minutes every 2 hours at a 2 Hz sampling frequency. Regarding the Datawell GPS buoy, De Vries et al. (2003) have shown that a single GPS receiver can achieve 1 cm precision for recording

wave periods up to 100 s. At the Lagos site, Shell Corporation collected conventional wave data in 2006 and 2007 at three locations using Nortek AWACs in 8, 12, and 15 m depths.

Lastly, the Naval Postgraduate School conducted an experiment in 2007 to observe wave evolution across the inner continental shelf at Martha's Vineyard in the North Atlantic Ocean. The experiment was funded by the Office of Naval Research Ripples Directed-Research Initiative, and the data were obtained from the National Oceanographic Partnership Program (NOPP) Shal-

low Water Repository website, when they were formerly available. Detailed descriptions of the location and experiment is





provided in Herbers and Janssen (2007). A magnetic declination for each site was obtained from the National Centers for Environmental Information (NCEI) website (NOAA NCEI) and accounted for in directional estimates.

## 3 Frequency spectrum analysis of infragravity waves

### 3.1 Methodology

This component of the study aims to build on the successful parameterisation approach adopted in the WAVEWATCH III (WW3) infragravity wave module, which is defined by Eq. (3) in Ardhuin et al. (2014). Here, we systematically examine the spectral properties of infragravity waves and their correlation with the wind-generated wave integrated parameters. This was undertaken with a view to further the WW3 infragravity wave height parameterisation in a dimensionless manner that is applicable to the nearshore coastal region.

Principal wave parameters for wave heights and periods are typically derived in the frequency domain from spectral analysis of the surface elevation signals. Accordingly:

$$H_{\mathrm{rms}} = 2\sqrt{2}\sqrt{m_0}, \; m_0 = \int_{fc,\mathrm{lo}}^{fc,\mathrm{hgh}} S_\eta(f)df, \tag{1}$$

$$T_p = \frac{1}{f_p}, \; \text{where } f_p \text{ is the frequency of } \max(S_\eta(f)), \text{ and} \tag{2}$$

$$T_{m02} = \sqrt{\frac{m_0}{m_2}}, \; m_n = \int_{fc,\mathrm{lo}}^{fc,\mathrm{hgh}} f^n S_\eta(f)df, \tag{3}$$

where $S_\eta(f)$ is the variance density spectrum, $H_{\mathrm{rms}}$ is the Root Mean Square (rms) wave height, $T_p$ is the peak wave period, and $T_{m02}$ is the second spectral moment wave period. The surface elevation, $\eta$, can be obtained from either direct or inferred measurements. Frequency separation between wind-generated and infragravity waves is denoted by $fc,\mathrm{lo}$, and the high fre-
quency cut-off is denoted by $fc,\mathrm{hgh}$. Categorically, an $fc,\mathrm{lo}$ value of 0.033 Hz is applicable even for a fully developed sea state, and an $fc,\mathrm{hgh}$ value of 0.5 Hz is generally considered sufficient to capture and characterise the wind-generated wave climate for the data used in this study. The $fc,\mathrm{hgh}$ value can also be limited by the nyquist frequency.

  Deriving an infragravity wave frequency spectrum with sufficient reliability differs from wind-generated waves in that the duration of a record length needs to be significantly longer to yield adequate resolution in the infragravity wave frequency
band. Typical sampling configurations to measure wind-generated waves comprise a wave burst ranging from 17 to 35 minutes in duration. Indeed, for 2 Hz in situ measurements, such as current meters with the waves mode, a 1,024 s (∼17 minutes) record length yields 2,048 data points suitable for the fast Fourier transform, while Datawell waverider buoy's on-board processing, with a sampling frequency of 1.28 Hz, are undertaken half hourly for $26\frac{2}{3}$ minutes of data. This is a commonly used method



to derive wave estimates and has been practiced over a number of decades. For infragravity wave parameters however, there is no established common practice, and their derivation requires attention in selecting a 'sufficient' record length. If we roughly take the longest peak period wind-generated wave expected in the oceans to be 20 s, then a 17 minute record length has approximately 50 samples, and we take this as a guide for selecting the required infragravity wave record length. A quandary

we face here is that the duration of a record length needs to be long to provide both sufficient resolution at very long periods relative to wind-generated waves and the adequate level of reliability, while still maintaining the stationary assumptions that are dependent on the sea state, tide, and the randomness of ocean surface motions forced by winds and atmospheric pressure. Accordingly, three hours is often taken as the upper limit duration that satisfies the steady sea state assumption, and we follow this practice. Considering this, and using 50 wave samples in a record length for the Fourier transform to be applied as a guide,

the longest peak period infragravity wave realistically resolvable is around 3 - 3.5 minutes (e.g., 50 samples $\times$ 200 s $\sim$ 2.7 h). Knowing that the ocean surface motions above around five minutes relate to atmospheric and other geophysical processes, this upper wave period limit seems acceptable. As such, the target duration of approximately 2.5 - 3 hour record length was used for the infragravity wave analysis. Infragravity wave estimates can be derived from the following expressions:

$$H_{\text{rms,ig}} = 2\sqrt{2}\sqrt{m_0}, \ m_0 = \int_{f\text{c,lo-ig}}^{0.033Hz} S_\eta(f)df, \text{ and} \tag{4}$$

$$T_{m02,\text{ig}} = \sqrt{\frac{m_0}{m_2}}, \ m_n = \int_{f\text{c,lo-ig}}^{0.033Hz} f^n S_\eta(f)df, \tag{5}$$

where $f$c,lo-ig is the low frequency cut-off, typically chosen as 0.0033 Hz, except for the Datawell GPS waverider buoy, which was governed by the instrument limit of 0.01 Hz.

Concerning sourced experimental data captured using the conventional wind-generated wave sampling regime (refer to

Table 1), the data were recorded in bursts considerably shorter than the target record length described above. For this type of data, we developed and applied a burst composition method where a number of wave bursts were aggregated in the frequency domain to increase the data length, such that a sufficient 'burst' duration can be captured. For example, three of the FRF AWAC data records with a single burst duration of approximately 34 minutes may be combined, such that more accurate infragravity wave estimates can be sought. The new 'synthetic' burst duration then equates to approximately 1.5 hour record length over

the 2.5 hour period. Although this is shorter than the target record length, the method allows us to examine infragravity waves from the conventional wind-generated wave data.

To study infragravity waves' correlative properties, we considered various wind-generated parameters. The fundamental concept of infragravity wave evolution is that the generation occurs during the wave transformation process from deep water to the shallow water. Commonly used dimensionless spectral parameters to capture the changing wave form include:

wave steepness $= ka$, and relative depth $= kd$, $\tag{6}$



where the wavenumber, $k$, follows the linear dispersion relation:

$$\omega^2 = gk\tanh(kd). \tag{7}$$

Furthermore, we know that wave grouping is also important to the infragravity wave generation. Wave groupiness is often inferred from spectral width parameters, and there are numerous expressions. We examined a number of spectral width parameters, and based on the applicability to this data set, we selected the one proposed by Cartwright and Longuet-Higgins (1956):

$$\epsilon_{\mathrm{sw}}{}^2 = 1 - \frac{m_2^2}{m_0 m_4} : 0 \leq \epsilon_{\mathrm{sw}} \leq 1. \tag{8}$$

The subscript 'sw' distinguishes this parameter from the $\epsilon$ symbol typically reserved for wave steepness.

At each site, we systematically conducted correlation analysis and examined the following density scatter plots:

- Infragravity wave height and wind-generated wave counterpart, $H_{\mathrm{rms}}$ vs $H_{\mathrm{rms,ig}}$;

- Wave height ratio, $\frac{H_{\mathrm{rms}}}{H_{\mathrm{rms,ig}}}$, vs wave steepness, $ka$;

- Wave height ratio, $\frac{H_{\mathrm{rms}}}{H_{\mathrm{rms,ig}}}$, vs inverse relative depth, $(kd)^{-1}$; and

- Wave height ratio, $\frac{H_{\mathrm{rms}}}{H_{\mathrm{rms,ig}}}$, vs spectral width parameter, $\epsilon_{\mathrm{sw}}$

The relative depth parameter is inverted for convenience, such that greater values represent shallower depths where the infragravity wave presence is prominent and a positive correlation with the wave height can be obtained. The wavenumber, $k$, for wind-generated waves was estimated with the linear dispersion relation, using the peak period measured at the deep water instrument location (e.g., the location where wind-generated wave data were collected) and translated to the depth where the infragravity wave measurements were made.

## 3.2 Correlation of infragravity waves to wind-generated waves

Considering the superior spatial data coverage for infragravity waves captured in the FRF AWAC data, a description of these data is presented to characterise the comparison of infragravity wave and wind-generated wave integrated parameters. Wind-generated wave estimates were made from the WRB-26m data, while infragravity frequency spectra were obtained from the AWAC5m data for all events (except Event 19, for which AWAC6m was used due to data availability). Infragravity wave parameters were also derived for all available AWAC data. Three hourly statistics were obtained for both waves; the wind-generated wave spectrum was produced by averaging six spectra, derived from half hourly records of 2,304 data points as stored in the FRF website (U.S. Army Corps of Engineers, FRF). Each half hourly wind-generated wave spectrum has 12 degrees of freedom at around $3.91 \times 10^{-3}$ Hz resolution. Regarding the infragravity wave spectrum, AWACs for these deployments were configured to collect wind-generated wave data in a series of wave bursts. As such, the data composition method described earlier was adopted. The infragravity wave spectrum was obtained by averaging three wave bursts in the frequency domain (i.e., equating to approximately a 1.5 hour record length spanning over three hours with 12,288 data points) to produce frequency



spectra with 42 degrees of freedom at around $1.95 \times 10^{-3}$ Hz resolution. The first resolvable frequency bin not including the zero frequency (i.e., mean of all frequencies) is centred at $2.93 \times 10^{-3}$ Hz.

We can examine the spectral properties and parameters of the recorded data in Fig. 3, which presents a compendium of wave climate information derived from the frequency spectrum. The unique aspect of these deployments is that a cross-shore

transect of infragravity wave estimates can be analysed. Comparing the spectrograms for both waves shown on the Fig. 3 top left panel, the energy magnitudes are correlated but the infragravity wave spectrum is much broader than the wind-generated waves. The top plot in the Fig. 3 bottom left panel depicts a wave height comparison, and it is interesting to note that the relative values of $H_{\mathrm{rms,ig}}$ and $H_{\mathrm{rms}}$ appear to vary considerably; the ratios of these heights can be as low as 0.1 but exceed 0.35, as it did in Event 2, when the longest peak period swell waves were detected. The bottom plot in the same panel presents the

infragravity wave heights from all the AWACs and clearly depicts a gradual increase or amplification for shallower depths. The magnitudes of infragravity wave height amplification at different depths are also not constant. For example, the relative values of AWAC5m and AWAC11m wave heights range between 1.2 and 2. A disparate pattern is observed for $T_{m02,\mathrm{ig}}$, as mean wave energy appears to shift to shorter periods for shallower depths as shown in the bottom plot of the Fig. 3 bottom right panel. The effect of peak wave periods to the infragravity wave response is pronounced when comparing Events 1 and 2, as smaller waves

with long periods in Event 2 generated similar magnitude infragravity waves to Event 1. Similar comparison characteristics have been observed for other sites.

Correlation analysis was undertaken to characterise the dependence of infragravity wave heights to wind-generated wave integrated parameters. Through an analysis of density scatter plot trends, we aimed to develop a simple parametric relationship of infragravity wave heights. Of particular interest, was to acquire an improved insight into the variability of the wave height

ratio, $\frac{H_{\mathrm{rms,ig}}}{H_{\mathrm{rms}}}$. Examples of scatter density plots are provided in Fig. 4, in which the colour scale gives the scatter density.

The infragravity wave theories and previous observations have shown that infragravity wave heights are strongly correlated with the wind-generated wave counterparts; this is consistent in the observational data as reflected by a Pearson's correlation r value of 0.81 for the data plotted in the top left panel of Fig. 4 and values ranging from 0.76 to 0.89 for all sites. With the objective of obtaining a relationship between infragravity wave heights and various wind-generated wave parameters, we

evaluate the wave height ratios against the inverse relative depth parameter - for example, see Fig. 4 bottom left panel. Overall, there is a positive dependence for most sites, with an average r value of 0.61. Although, this mean value is dampened primarily by the lack of a strong correlation from the Baja and Lagos data, with r values of 0.29 and 0.53, respectively. The upper bounds of the scatter envelope at these two sites is consistent with those observed at the other sites. In an exploratory manner, we calculated a wavenumber based on the mean wave period, $T_{m02}$, and substituted this into the inverse relative depth parameter

calculation. This improved the correlation with increased r values of 0.64 and 0.73, respectively. The likely explanation of the improved dependence, when the wavenumbers were derived from $T_{m02}$, is a noisy fluctuation of the peak wave period (i.e., the peak wave period solely was not a stable representation of the dominant wave period characteristics). The significance of this correlation is that the relationship is in a dimensionless form and accounts for a number of principal variables, including the wave height, period (indirectly through the wavenumber), and the depth where the infragravity wave measurements were

made.



We also note that there is a strong correlation between the wave height ratios and the spectral width parameter, with r values ranging from 0.64 to 0.82. From this, we can infer that for a given wind-generated wave height, increased groupiness can manifest in a greater infragravity wave response, which is consistent with the infragravity wave theory. There is also evidence of nonlinear behaviour of the wave height ratios for very narrow spectral width, e.g., $\epsilon_{sw} > 0.80$, which is pronounced in the

Lagos data (not shown here). Regarding the wave steepness, no conspicuous trend was observed in the correlation analysis.

### 3.3 Correlation analysis interpretation

As described earlier, the peak period can be a noisy parameter, and can also be dependent on sampling configurations, spectral analysis setup and methods, and the characteristics of measured wave systems. Scatter is apparent in all density scatter plots of the wave height ratios and the inverse relative depth parameter. Aiming to minimise the effect of noisiness, we smoothed the

peak wave period parameter by adopting a simple moving average filter of 20 hours, so that the influence of more variable wind seas can be removed. This improved the correlation between the wave height ratios and the relative depth parameter at all sites. The average correlation value from all data increased from 0.61 to 0.67. This and the categorical increase of the correlation demonstrate that the scatter is attributable, at least part there of, to the noisiness of the peak wave periods.

A strong correlation between the wave height ratios and the spectral width parameter has been observed, suggesting that

the narrow spectral width, or the groupier incident wind-generated wave, is associated with an increased infragravity wave response, which conforms to infragravity wave theory. We simplistically incorporated the spectral width parameter by multiplication with the inverse relative depth parameter, which also results in a marginal categorical increase of the correlation coefficients, with an average r value of 0.75.

Figure 5 is a scatter plot of the infragravity wave to wind-generated wave height ratios against the product of the inverse

relative depth and the spectral width parameter for all data. Two biases to consider when assessing this plot are the disparate ranges and the sample populations, which differ considerably at each site. Specifically, the FRF AWAC data have a large range but lack the sample population, whereas the FRF pressure array and Lagos data have a sample population nearly ten times the FRF AWAC data but their ranges are limited. The Baja and Martha's Vineyard data have both limited sample populations and ranges due to quiescent sea states in the data. The FRF pressure array and Lagos data in Fig. 5 have similar result ranges and

sample populations. The scatter in the Lagos data is significantly greater than the FRF pressure array data. Further, comparing both data sets at the FRF site, the AWAC data have greater scatter than the pressure array data. The degree of the scatter appears to coincide with the record lengths of the infragravity wave samples used for their corresponding spectral analysis, which were (the wind-generated wave record length in the brackets) 68 (17), 102 (34), and 170 (170) minutes for the Lagos, FRF AWAC, and FRF pressure array data, respectively. The scatter observed in our data may be partly attributable to the sampling

variability effect, particularly those wave data that were collected using the conventional sampling configurations to measure wind-generated waves. Bitner-Gregersen and Magnusson (2014) showed that the sampling variability, which is dependent on the sample length, can also increase with increasing wave heights and periods. This is a conundrum that is difficult to overcome when the data were opportunistically sourced, especially considering the lack of field-dedicated infragravity wave



measurements readily available. It is however encouraging to still obtain the infragravity wave height dependence to the relative depth parameter, $kd$, with a correlation coefficient value of around 0.67 for the total data period exceeding two years.

The nonlinearity observed for wave height ratios with the very narrow spectral width perhaps suggests there is more work required to consolidate its correlation with the infragravity waves. But the strength of the correlation with the inverse relative

depth parameter supports the application of a simple linear regression approach. Figure 6 presents infragravity wave height estimates from the regression model against the measured counterparts for all valid data used in this study. A correlation coefficient value of 0.93 provides an optimistic basis for the formulation of a new dimensionless parameterisation of infragravity wave heights at nearshore sites.

Other parameters and physics not accounted for in this experimental analysis include:

– An estimation of bound wave energy: there are numerous studies, which demonstrate that the statistics related to the bispectrum can be used to estimate the bound wave energy (Herbers et al., 1994; Reniers et al., 2002), however, the statistical uncertainty in the bispectral estimates is relatively large when the nonlinearity is weak (Herbers et al., 1994), and the method becomes unreliable for high waves when the assumption of a slowly varying wave group is no longer valid (Reniers et al., 2002, 2010). Accordingly, we have focused on the total infragravity wave heights and their dependence

to the wind-generated wave integrated parameters.

– Effects of bed slope: a carefully designed field experiment, in which data at two sites with similar deep water wave climate (at least in magnitudes) with contrasting bed slopes, would be required to investigate bed slope effects. Wave steepness can be a function of the bed slope, however, the wave steepness by itself did not show any correlation with the infragravity wave heights in our analysis.

– Effects of infragravity wave dissipation at the shoreline and reflected wave energy: this is somewhat related to the previous point, as it is envisaged that reflection coefficients vary between different geographical settings, e.g., surf zone slopes. Again, a carefully constructed field measurement program would be needed to account for this process in the field.

## 4   Directional analysis of infragravity waves

### 4.1   Methodology

An experimental investigation to examine the directional properties of infragravity waves was conducted using conventional stochastic directional analysis methods. A fundamental thesis to address here begins with an elementary concept, which is the feasibility of applying these methods to obtain directional estimates for infragravity wave frequencies, specifically the Maximum Likelihood Method (MLM) and the Maximum Entropy Method (MEM). The feasibility is assessed based on a qual-

itative assessment of the derived directional distributions of the infragravity waves. We also assess the possibility of deriving quantitative parameters, such as mean directions and directional spreading.





There are primarily two types of measurement methods to capture the directional properties of waves; a point measurement in which three physical quantities are captured or a spatial array of wave gauges in which a number of measured quantities depends on the number of sensors. The number of measured quantities are generally far fewer than those suggested in Young (1994) to adequately describe the two-dimensional wave form. Consequently, stochastic methods based on the Fourier coefficients are

the most widely adopted approach at the present-day, including the MLM and the MEM, among others. These two directional analysis methods were used for this study, and their expressions are concisely described, closely following Young (1994) and Benoît et al. (1997).

Frequency-directional spectra of ocean waves are commonly expressed in the form:

$$S(f,\theta) = E(f)D(f,\theta), \tag{9}$$

where $E(f)$ is the one-sided variance spectrum and $D(f,\theta)$ is the Directional Spreading Function (DSF). A DSF follows:

$$D(f,\theta) \geq 0 \text{ over } [0,2\pi], \quad \text{and} \quad \int_{0}^{2\pi} D(f,\theta)d\theta = 1. \tag{10}$$

Further, it is also common to express a DSF as a function of radian frequency, $\omega$, as well as a wavenumber. The wavenumber and direction $(k,\theta)$ follow:

$$S(f,\theta) = \frac{2\pi}{C_g}S(k,\theta) = \frac{2\pi k}{C_g}S(k_x,k_y), \tag{11}$$

where $C_g$ is the group velocity for frequency, $f$, based on the linear dispersion. The commonly represented time and space variant water surface elevation, $\eta$, is assumed as:

$$\eta(x,y,t) = \iint \sqrt{2S(f,\theta)}dfd\theta \, \cos[k(x\cos\theta + y\sin\theta) - 2\pi ft + \psi], \tag{12}$$

given phase function $\psi$ is randomly distributed over $[0, 2\pi]$ and there are no phase-locked waves (e.g., waves reflected in the vicinity of a few wavelengths). If various components of the wave field are randomly distributed, the following equation is

obtained between the DSF and the cross spectra, $G_{mn}$, of any two measured quantities m and n:

$$G_{mn}(f) = \int_{0}^{2\pi} T_m(f,\theta)T_n^*(f,\theta)e^{-ik(x_n-x_m)}D(f,\theta)d\theta, \tag{13}$$

where $D(f,\theta)$ is the decomposition of Equation (9) and $T_x(f, \theta)$ is the transfer function used to obtain equivalent surface elevation from indirect measured quantities (e.g., pressure, velocity etc.). Young (1994) and Benoît et al. (1997) provide a comprehensive list of transfer functions. The symbol * stands for the conjugate operator as $T_x(f,\theta)$ is a complex function in

the general case.

Lygre and Krogstad (1986) have shown that estimates of $D(\theta)$ at a particular frequency, $f$, using the MEM can be found based on the primary Fourier coefficients:

$$2\pi D(\theta) = \frac{(1 - \xi_1 c_1^* - \xi_2 c_2^*)}{|1 - \xi_1 e^{-i\theta} - \xi_2 e^{-2i\theta}|^2}, \tag{14}$$





using the following identities:

$$c_1 = a_1 + b_1,$$

$$c_2 = a_2 + b_2,$$

$$\xi_1 = \frac{(c_1 - c_2 c_1^*)}{(1 - |c_1|^2)}, \text{ and}$$

$$\xi_2 = c_2 - c_1 \xi_1,$$

and the Fourier coefficients at each frequency can be derived from a DSF using:

$$\text{DSF}(f,\theta) = \frac{a_0}{2\pi} + \frac{1}{\pi} \sum_{n=1}^{\infty} [a_n \cos(n\theta) + b_n \sin(n\theta)], \tag{15}$$

where

$$a_0 = \int_0^{2\pi} D(f,\theta) d\theta = 1, \tag{16a}$$

$$a_n = \int_0^{2\pi} D(f,\theta) \cos(n\theta) d\theta, \text{ and } b_n = \int_0^{2\pi} D(f,\theta) \sin(n\theta) d\theta. \tag{16b}$$

Directional estimates for data collected via a point measurement were derived based on the above expressions.

A spatial array of wave gauges captures phase lags between each pair of sensors, and the MLM can be applied to obtain directional estimates from spatial data. Following Reniers et al. (2010) (with reference to Davis and Regier (1977) and Pawka (1983)), a wavenumber directional spectrum for a given frequency, $f$, can be estimated based on:

$$D(k,\theta) = \kappa \Big[ \sum_{n=1}^{N} \sum_{m=1}^{N} G_{mn}^{-1}(f) \exp^{-(ik\sin\theta(y_m - y_n) + k\cos\theta(x_m - x_n))} \Big], \tag{17}$$

where $x_m$ and $y_m$ correspond to the cross-shore and alongshore location of the sensor 'm' within the array. $G_{mn}$ is the cross spectra of 'm' and 'n' sensors, and $\kappa$ is the normalization factor such that the total energy of the frequency-directional spectrum and the frequency spectrum are equivalent. Integrating the wavenumber spectrum yields a frequency-directional spectrum, $E(f,\theta)$. This method was applied to the array of pressure sensors at the FRF site to examine the directional properties. Further, the directional spectrum modification based on an iterative process described in Pawka (1983) was adopted. 10 iterations were found to be sufficient to resolve directional peaks in the observations and applied in this study.

The feasibility of directional analysis methods and the rationality of derived infragravity wave directional estimates were assessed by comparing directional spectrograms of both the wind-generated and infragravity waves. A directional distribution of the peak frequency bin can be considered to represent the wind-generated wave directional characteristics, while the mean directional distribution, obtained from an integral of the directional spectrum over the infragravity wave frequencies, was used to represent the infragravity waves. A mean distribution was selected for the infragravity waves, as the bottom plot of





the Fig. 3 top left panel, depicting their distribution in frequency, shows that their peaks are often poorly resolved. It has therefore been inferred that the directional distribution of the peak infragravity frequency alone has insufficient accuracy to provide reliable estimates of the infragravity wave directional properties. The infragravity wave directions would be expected to exhibit some form of dependence to the wind-generated wave counterparts, as well as demonstrate a degree of conformance

to the geographical characteristics of measurement sites, such as a coastline orientation. We rely on these ideas to qualitatively assess the validity of the derived infragravity wave directional distributions. Directional spectrogram figures (for example, Fig. 7) are supplemented with deep water wind-generated wave peak directions, $\theta_{\text{peak}}$, depicted in a white line to visually aid in the incident wave propagation comparison.

Interpretation of wave directional properties from integrated parameters can often be complicated. The principal parameters

that are generally used to describe their properties include a peak direction, $\theta_{\text{peak}}$, and a mean direction, $\theta_{\text{mean}}$, and they are calculated based on the following expressions:

$$\theta_{\text{peak}} = \arctan \frac{\int_{-\frac{\pi}{2}}^{\frac{\pi}{2}} (uu)d\theta}{\int_{-\frac{\pi}{2}}^{\frac{\pi}{2}} (vv)d\theta} \text{ of the } f_p \text{ bin, and} \tag{18}$$

$$\theta_{\text{mean}} = \arctan \frac{\int_{-\frac{\pi}{2}}^{\frac{\pi}{2}} \int_{fc,\text{lo}}^{fc,\text{hgh}} (uu)dfd\theta}{\int_{-\frac{\pi}{2}}^{\frac{\pi}{2}} \int_{fc,\text{lo}}^{fc,\text{hgh}} (vv)dfd\theta}, \tag{19}$$

where $uu$ and $vv$ are the Cartesian expression of $D(f,\theta)$, and $-\frac{\pi}{2}$ and $\frac{\pi}{2}$ represent the wave propagation sector. While these parameters describe the dominant propagation directions of waves, a distribution of energy over the wave propagation sector is often expressed in the form of a directional spreading parameter. Conventionally, a parametric approach is adopted to describe the spreading. The parametric functions can vary in complexity, from an elementary form of cosine power n, and cosine to the power 2s (Longuet-Higgins et al., 1963), to a representation of spreading as a function of non-dimensional frequency, $\frac{f}{f_p}$,

as well as accounting for the wave age (Hasselmann et al., 1980; Donelan et al., 1985). A concise review of these spreading functions are described in Sect. 2 of Young (1994). More recently, Ewans (1998) developed a parametric function that describes the bimodal spreading observed in wind seas. Describing the directional spreading in this manner inherently assumes that a derived directional spectrum can be fitted to these parametric functions. For experimental infragravity wave data, which are expected to be broad and multi-modal based on the limited literature, a more robust approach may be suitable; Babanin and

Soloviev (1998) developed an 'A' spreading parameter, which is simply an inverse of a directional spectrum integral expressed as follows:

$$'A'^{-1} = \int_{-\frac{\pi}{2}}^{\frac{\pi}{2}} D(\theta)d\theta, \tag{20}$$

where $max(D(\theta))$ is normalised to unity, allowing for a comparison of spreading of waves with different magnitudes. For clarity, large 'A' values represent narrow directional distributions. 'A' parameter's interdependencies with other conventional

spreading parameters, such as cosine to the power n, 2s (Longuet-Higgins et al., 1963), and $\beta$ (Donelan et al., 1985) are provided in Fig. 3 of Babanin and Soloviev (1998).



## 4.2 Qualitative analysis

The objective of the analysis in this section was to acquire improved insights into the infragravity wave directional properties, in particular, their relationship with the wind-generated wave counterparts. Similar to Sect. 3, the FRF AWAC data are described in detail. These data contain a transect of shallow water measurements and facilitated observation of changes in the infragravity wave directions at various depths. The site was also considered the most ideal location to assess the feasibility of the adopted directional methods, as a series of concurrent coherent directional estimates would enhance confidence in the validity of the derived directional distributions, at least qualitatively.

A series of infragravity wave directional spectrograms was plotted in Fig. 7 for all AWAC data, as well as the peak frequency and mean (frequency integrated) directional spectra of the wind-generated waves. In addition to the white line, which depicts the peak wind-generated wave directions, dashed magenta lines were plotted in these figures to represent a shore-normal orientation of the data site. The directional plots in Fig. 7 appear coherent. Notable features of the observed infragravity wave directional physics include:

- Infragravity wave directional spreading is well-defined during the episodes of significant energy. Further, overall characteristics of the derived directional spectrum from all four different AWACs are consistent.

- Directional distributions are often bimodal with directional peaks in both incident and reflected sectors. In particular, a distinct bimodal directional structure emerges during the episodes of energetic waves. The bimodal nature of the infragravity wave directional distributions have support from the infragravity wave theory, -viz. infragravity waves develop through the amplification of incident waves in the shallow water, and they are reflected from the coastline to the deep ocean. This also means that appropriate directional discretisation is required to accurately represent the infragravity wave parameters (i.e., a discretisation of wave heights into incident and reflected components).

- During the calm periods, depicted in contour colours below yellow in the infragravity wave directional spectrograms, the infragravity wave directional distributions appear complicated - the directional distributions are broad and smeared with a large portion of energy spread over the oblique propagation directions, perhaps indicating the presence of dominant edge waves.

- Infragravity wave directional spreading is generally broader than the wind-generated wave counterparts.

- Focusing on the incident propagation of the infragravity waves during the energetic episodes, there is a correlation between the wind-generated wave peak directions, $\theta_{\mathrm{peak}}$, and the infragravity wave directional peaks. This is consistent for all AWACs when distinguishable peaks exist. We also observe that when the deep water peak directions are oblique (e.g., towards southeast), infragravity waves appear to refract and propagate in a shore-normal orientation in shallower depths. This feature is prominent in Events 2 and 7.





- Given that the infragravity waves propagating in the incident directions appear coherent, and if we assume that reflected infragravity waves generally obey the laws of specular reflection, then the outgoing infragravity waves also appear to propagate in coherent directions with respect to their incident waves.

During the quiescent periods, for example, at the end of Events 1 and 19, it is clear that the directional distributions are broad and smeared with a large portion of energy spread over the oblique propagation directions. It is interesting to note however, that when comparing Events 1 to 3 from deeper AWACs to the shallower instruments, the emergence of incident and reflected peaks from the broad smeared distributions can be observed. From this, we can infer that in the absence of dominant directional peaks, edge waves can often dominate, while a more simple directional structure is formed as the infragravity energy becomes increasingly prominent in the shallow water. From a qualitative perspective, the directional estimates derived using the MEM conform to the theories of infragravity waves and the geographical setting at the field site. Similar qualitative analysis findings were obtained for the Lagos AWAC data and described in Nose et al. (2016).

Concerning the FRF pressure array data, the plots in Fig. 9 depict the monthly segmented directional spectrograms for infragravity waves. Compared to the wind-generated waves in Fig. 8, their distributions are so broad that their peaks are difficult to ascertain. When the peaks are distinct, they appear either in the directional sector incident to the coast or smeared over oblique propagation directions, i.e., between west and east. Reniers et al. (2010) interpreted the latter as edge wave dominated events. The log to the base 10 colour scale has insufficient resolution to distinguish the dominant infragravity wave directions, and as such, normalised directional spectrograms for two events in January and April were plotted on the bottom panels in Fig. 10, while the top panels show the wind-generated wave directional spectra. The following observations can be made:

- In the normalised directional spectra, there are definitive peaks strongly related to the white dashed line, which depicts the peak wind-generated wave directions, $\theta_{\mathrm{peak}}$, in deeper water. The bottom panel in Fig. 10a contains a back and forth transition of the peak wind-generated wave directions between the northeast and southeast sectors. The positive association with the incident infragravity waves is still observed during the transient wave event, demonstrating the robustness of this trend.

- The infragravity wave spectra on the bottom panels have directional peaks apparent in both incident and reflected sectors. The dominant directions generally appear more prominent in the propagation sector incident to the coast.

- Infragravity wave directions are consistently multi-directional and complex. In these example directional spectra, we can broadly characterise that for peak directions, $\theta_{\mathrm{peak}}$, between east and southeast, bimodal directional distributions occur in the infragravity waves and oblique incident directions results in the multi-modal peaks.

Amidst these complex directional spectra, the dependence of the incident infragravity wave directions to wind-generated waves is evident. The persistent dominant incident directions of infragravity waves in the observations are somewhat unexpected. Previous studies have generally only attributed 20 - 30 % of total infragravity wave energy to the bound wave components (Herbers et al., 1994; Reniers et al., 2010; Ardhuin et al., 2014). Reniers et al. (2002) estimated that the bound infragravity





wave energy contribution stays less than 50 % throughout the surf zone. The dominant incident directions suggest a potential for the presence of free incident propagating infragravity waves. The multi-modal distribution is likely associated with edge waves, but an accurate account of their propagation is difficult to resolve, primarily due to the complexity of the associated directional distributions. The FRF pressure array data were specifically designed to measure the directional properties of infragravity waves. Applying the conventional MLM directional analysis method to the high spatial resolution array of pressure sensors provides evidence that qualitative information of infragravity wave directions can be sought during energetic wave events. However, the multi-modal distributions of the derived spectra indicate the infragravity wave directionality is extremely complex.

Due to a combination of small infragravity wave signals and the limited data duration of statistically significant wave events at the Baja and Martha's Vineyard sites, we were unable to build a site specific profile of directional characteristics, to assess the validity of the derived directional spectra, for these locations.

### 4.3 Quantitative analysis

Through the directional observations at each data site, it has been found that comprehensible directional distributions emerge when energetic signals were detected, which are potentially useful to carry out a quantitative investigation. The presence of dominant edge waves or quiescent conditions result in the derived directional distributions appearing too complicated for inter­pretation and analysis in a quantitative manner. As such, selection criteria to remove the complex distributions were developed. The selection criteria are described in Nose et al. (2016) and were based on the directional peak prominence to determine whether the derived directional spectra are useful for the purposes of quantitative analyses. The selection procedure focused on identifying the presence of a dominant infragravity wave system with unique peaks in the incident and reflected directional sectors. The assessment of the directional data using the selection criteria found that infragravity waves often comprise multi­ple directional peaks and at least one peak in each of the incident and reflected sector. Consistent with the qualitative analysis, it was shown that when infragravity wave energy increased, their directional distributions become predominantly bimodal, having unique prominent peaks in both incident and reflected directions. Indeed, more than 80% of the filtered AWAC data at the FRF and Lagos sites have the bimodal directionality for infragravity wave heights, $H_{\mathrm{rms,ig}}$, greater than 0.1 m. The FRF pressure array data depict an entirely different trend; they mostly comprise a unimodal spectrum, and the selection algorithm identified approximately 100 bimodal spectra for the entire data period, less than 5 % of the total number of observations. Although the presence of directional peaks in both sectors is apparent, the peaks in the opposing sector are not sufficiently prominent. The FRF pressure array data comprise a small number of the bimodal spectrum as their directional distributions are generally complex, and their usability were restricted to qualitative analysis in this study. Regarding the filtered FRF AWAC data with the bimodal spectrum, Fig. 11 presents the dominant propagation of the wind-generated and infragravity waves, derived from Eq. (18) and (19), respectively. It is evident in this figure that when unique prominent peaks are detected, mean propagation directions, $\theta_{\mathrm{mean}}$, for infragravity waves can be reliably derived. This is an encouraging outcome of the infragravity wave directional analysis.





The qualitative directional analysis has demonstrated that the infragravity wave spreading is significantly broader than the wind-generated wave counterparts. Here, we quantitatively examine the infragravity wave directional spreading estimates. The value of a specific estimate of directional spreading depends on the analysis methods. Spreading parameters, such as cosine to the power of n and 'A' therefore provide a relative measure only; for example, directional spreading derived from the same

data using the MLM and the MEM would result in different numerical values. The directional spreading for the FRF AWAC and Lagos data are plotted in Fig. 12. For wind-generated waves, the mean and peak frequency spreading was derived and arbitrarily factored, such that they are comparable to the infragravity wave spreading for plotting purposes. Although limited in duration, Fig. 12a and 12b provide evidence that the directional spreading of the wind-generated and infragravity waves are correlated. The directional spreading for the Lagos data comprise several instances when narrow spreading is apparent in both

waves. For both sites, two distinct periods between 20 and 40 wave counts in Fig. 12a (which corresponds to Event 2) and around the wave count 400 in Fig. 12b for the FRF AWAC and Lagos data, respectively, depict a definitive positive correlation. These figures also demonstrate the lack of consistency in the results, and more detailed correlation analysis is unlikely to provide improved insights; it is not clear whether this is due to insufficient resolutions of the measurements and/or the analysis methodology. But the limited observations of correlated directional spreading between the two waves is valuable information.

## 4.4    Numerical verification of the observations

Infragravity wave modelling of the analysed experimental field data was conducted using IDSB, a linearised 1D surfbeat model developed by Reniers et al. (2002), for comparison with the measurements. IDSB simulates infragravity waves in the coastal region based on the prescribed wind-generated wave spectral parameters at the offshore boundary and accounting for bottom friction, surface water level setup, and rollers. IDSB is limited to resolving the timescale of the wind-generated wave

groups for alongshore uniform bathymetry, but its governing equations are in the time domain, so the model can produce two dimensional surface elevation at a given point to derive directional estimates as described in Reniers et al. (2010). The model has been validated to reproduce bulk spectral statistics, in particular, wave heights, at the FRF site including data described in Birkemeier et al. (1997) and Reniers et al. (2002, 2010). In this study, the AWAC data also at the FRF site were used; the site is suitable for the numerical modelling applications, as it comprises a straight open coastline with relatively uncomplicated

bathymetry, and the collected AWAC data provide the most tenable outcome in the infragravity wave directional analysis.

The model adopted a simplified cross-shore profile between the shoreline and the location of WRB-16m that is consistent with the bathymetry in NOAA Electronic Navigational Chart 12204. The model output points closely followed the instrument locations and were provided at the AWAC depths of 5, 8, and 11 m, labelled PTP_5m, 8m_Array, and PTP_11m, respectively. AWAC6m was conglomerated with AWAC5m, as we are primarily interested in the qualitative comparison of directional

distributions. An additional output point, PTP_2m, at the 2 m depth grid was afforded to gain an understanding of the shallow water directional behaviour.

The model forcing was derived from three hourly spectral parameters from the WRB-16m frequency-directional spectra readily available on the FRF archive site (U.S. Army Corps of Engineers, FRF). Effects of surface water level fluctuations due to tides were not included in the simulations. According to the literature, there are differing effects of tides to the predictability





of infragravity waves and that these are site specific. For example, Thomson et al. (2006) observed a strong tidal modulation of infragravity waves due to nonlinear energy transfers in the surf zone along several kilometres of the southern California coast. On the contrary, Bijl et al. (2009) evaluated the accuracy of IDSB model estimates at the Baja and FRF locations in the 20 and 8 m depths of water, respectively. Their findings determined that the inclusion of tidal water level fluctuations led to little

differences in the resultant estimates of infragravity wave heights. Despite these disparate effects, it is reasonable to expect that including surface water level fluctuations would provide more robust estimates of infragravity waves. However, our objective of this investigation was to gain insights into model's directional predictions, which were not expected to be significantly affected by small changes in the still water level. As such, tidal surface water level variations were not included in this exercise.

In IDSB, a model grid is initially generated based on the user prescribed number of grid cells, however, the coarsest grid

resolution that is tolerated in the model has a threshold value of 4 m, and the model autonomously modifies and assigns the number of grid cells accordingly. It also requires a minimum computational depth as an input, and a value of 0.1 m was assigned, yielding a cross-shore distance of the 'wet' simulated model domain of approximately 3,410 m. Applying the 4 m resolution threshold, the model domain comprised 876 grid cells with around 3.9 m horizontal resolution. A sub-function named, *mwavetrans.m*, in IDSB calculates the mean wave transformation parameters at each grid, which are dependent on

three key IDSB parameters: the wave breaking saturation parameter ($\gamma$), the wave breaking parameter ($\alpha$), and the dissipation parameter (*ndis*). For the simulations with a simplified model setup, default values of 0.45, 2, and 5 were assigned, respectively. Since the IDSB model was developed and validated based on the measurements at the FRF site, these default values are envisaged to provide reasonable first estimates. We can also define the desired frequency spectrum resolution and 0.005 Hz was specified.

First, we assess the wave heights to ensure that the IDSB model is reproducing the observed AWAC data events to a reasonable approximation. As there were no data from the AWAC5m instrument during Event 19, the data from AWAC6m and the modeled wave heights at 6 m depth were included in Fig. 13 under the '5m' label. Notwithstanding the overestimation for waves greater than around 0.5 m, Fig. 13 depicts that there is a good agreement between the modelled and measured infragravity wave heights. The skill of the IDSB predictions is estimated using the skill parameter defined by Gallagher et al. (1998):

$$skill = 1 - \frac{\sqrt{(H_{\text{ig,measured}} - H_{\text{ig,modelled}})^2}}{\sqrt{H_{\text{ig,measured}}^2}}. \tag{21}$$

Based on this, calculated IDSB skill values range between 0.70 and 0.80 for three model output locations related to the AWACs. Considering that minimal tuning of the model input parameters were carried out and the tidal fluctuations were not included, the skill values are acceptable and demonstrate the robustness of the IDSB model as expected from Bijl et al. (2009) and

Reniers et al. (2010).

The MLM was applied to the surface elevation at all output points, and Fig. 14 presents the derived directional spectrograms. The directional properties derived in IDSB qualitatively exhibit a striking resemblance to Fig .7, which is a depiction of the measured infragravity wave directional spectrum. Consistent features in the simulated results include a highly correlated propagation of the wind-generated waves and the incident infragravity wave directions, and the specular behaviour of outgoing



waves from the coastline. Furthermore, a complex directional structure is prevalent during the quiescent conditions, another feature that is consistent with the in situ observations. A shore normal tendency of the infragravity waves can be seen, which becomes pronounced in the shallowest output point at PTP_2m. The dominant energy in the model is distributed in the directional sector outward from the coast, which is in contrast to the observations, however, agrees with the previously derived

IDSB directional spectrograms shown in Fig. 3 of Reniers et al. (2010). A slight overestimation bias for outgoing waves is noted in Reniers et al. (2010), and perhaps, the disparity can be partly attributed to this. Notwithstanding the disparity, the derived directional distributions in Fig. 14 reaffirm the infragravity wave directionality from the in situ observations.

## 5  Summary

We have examined characteristics of the infragravity wave field at nearshore sites on the west and east coasts of America and

the west coast of Africa. Our focus was directed at the directionality of the wave field, which has not received much attention in the past. Measured data available to us for this study amounted to a total duration exceeding two years. The data were measured using pressure sensors, Doppler sonar instruments, and waverider buoys and collected at the FRF, Martha's Vineyard, and Baja sites on the American coasts, as well as Lagos, Nigeria. The results of our analysis of these data have given improved insights into the spectral and directional properties of infragravity waves.

In accordance with earlier findings and theory, we found a strong dependence of the infragravity wave height on the incident wind-generated wave heights. A correlation coefficient of 0.67 was found between the ratio of the infragravity wave heights to the wind-generated wave height against the relative depth parameter, $kd$. The strong correlation supported the application of a simple linear regression approach, which enabled estimates of infragravity wave heights that agreed with the observations, with a correlation coefficient of 0.93.

We estimated infragravity wave directional spectra using conventional MLM and MEM analysis methods, and compared them with the corresponding directional spectra of the wind-generated waves, with the following observations:

- The derived directional distributions are invariably multi-modal, with peaks in incident and reflected sectors.

- When there is a dominant infragravity wave system detected in the data, the infragravity wave directional distributions are usually bimodal with the dominant energy in the propagation sector incident to the coast.

- There is generally a strong positive association between the peak wind-generated wave directions and the incident infragravity wave directions, but the incident infragravity wave directions in the shallow water depths tend to propagate in more of a shore-normal orientation compared to the wind-generated waves when their propagation is oblique.

- Outgoing waves appear consistent with the respective incident propagating waves, being specularly reflected from the coastline, as they propagate to the deep ocean.

Despite these observations, the infragravity wave directional properties are complicated and not all of the derived directional distributions appear to be sufficiently resolved. Accordingly, selection criteria were applied to identify events that could be

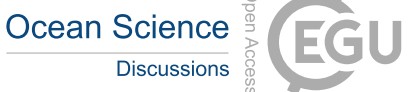

clearly related to the wind-generated waves, and these led to limited but definitive evidence of a directional spreading corre-lation between the infragravity and wind-generated waves – viz. the spreading of the infragravity waves was lower when the spreading of the wind-generated waves was narrower. In addition, the mean propagation direction of the infragravity waves could be determined reliably for the selected data at the FRF site.

5    General agreement between the results of the field data analyses and predictions with the IDSB infragravity wave model was found. IDSB simulations of the storm events for which FRF AWAC data were available was undertaken and compared to the observations. The IDSB model produced comparable total wave height estimates and the directional estimates were qualitatively consistent with the infragravity wave directional observations.

*Data availability.* The FRF AWAC data are available on the U.S. Army Corps of Engineers, Coastal and Hydraulics Laboratory, Field
10    Research Facility website provided in References.

*Competing interests.* The authors declare that they have no conflict of interest.

*Acknowledgements.* This work has been completed as part of the Ph.D. project of Takehiko Nose, titled "Infragravity Waves, Observations and Modelling", and was fully funded by Shell Corporation. The authors sincerely thank Dr Octavio Sequeiros and Shell Corporation for their generous support of this research work.
15    AVB acknowledges support from the DISI Australia-China Centre through grant ACSRF48199 and from the Australian Research Council Discovery Grant DP170100851.



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



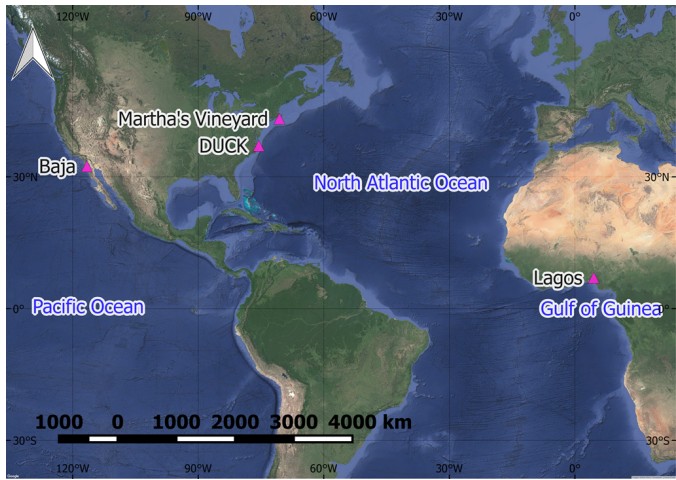

**Figure 1.** Location map of data sites.

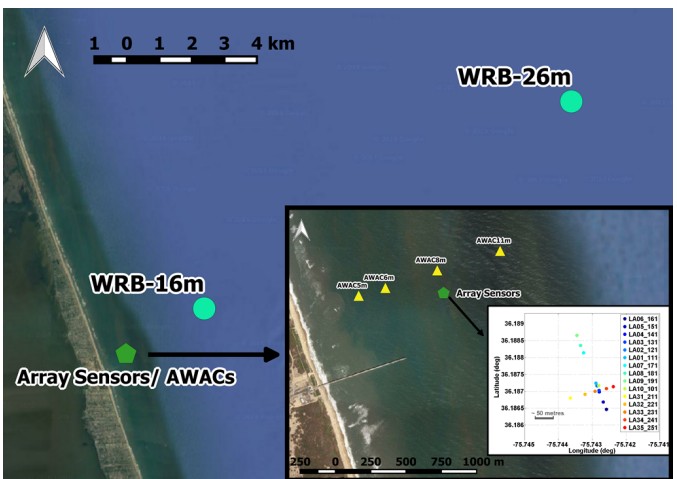

**Figure 2.** Location map of FRF instruments.

**Table 1.** Overview of experimental data sites.

| Location | Ocean | Approx. duration | Approx. IG wave gauge depth (m) | Targeted IG data collection |
|---|---|---|---|---|
| Duck, North Carolina | North Atlantic | 9 months | 8 m | Yes |
| Baja, California | Pacific | 4 months | 20 m | Yes |
| Lagos, Nigeria | Gulf of Guinea | 8 months | 8 m | No |
| Martha's Vineyard, Massachusetts | North Atlantic | 2 months | 10 m | No |
| Duck, North Carolina | North Atlantic | 28 days | 5, 6, 8, & 11 m | No |



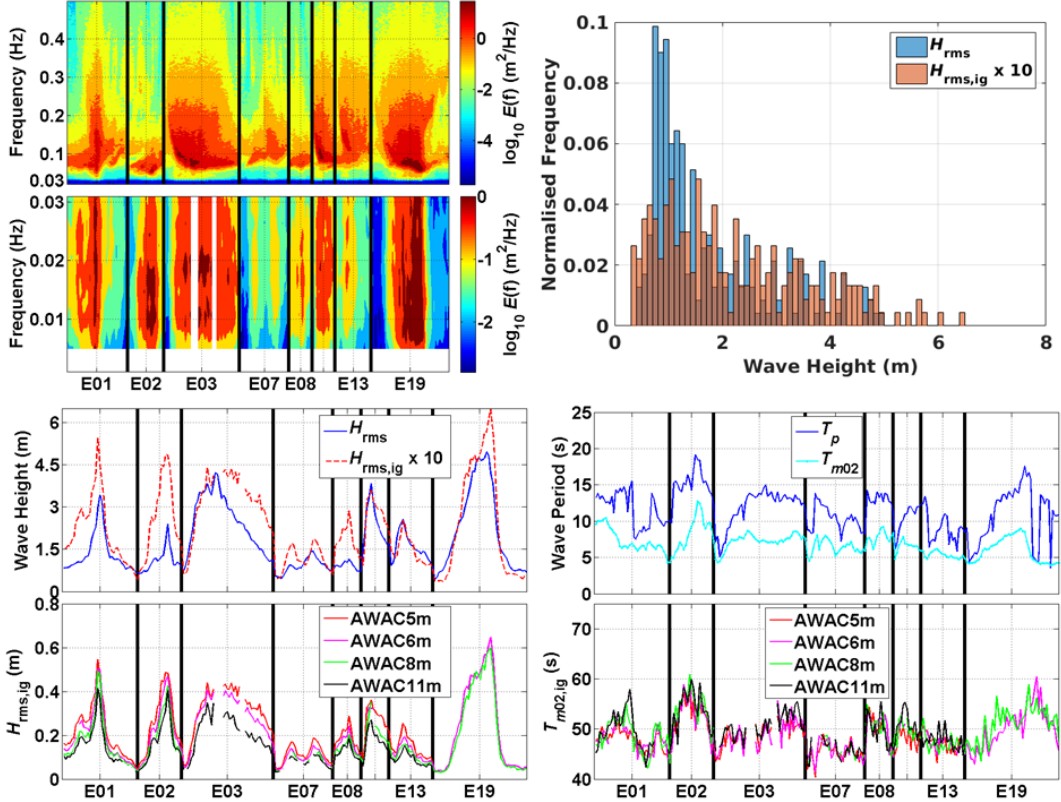

**Figure 3.** Overview of wave climate at FRF site, AWAC data: Wave spectrograms (top left), wave height frequency histogram (top right), wave height time series (bottom left), and wave period time series (bottom right).

**Table 2.** Availability and duration of storm archive wave data at FRF site.

|  | AWAC 5m | AWAC 6m | AWAC 8m | AWAC 11m | WRB-24m | Start date (dd/mm/yyyy) | Duration (days, hours) |
|---|---|---|---|---|---|---|---|
| Event 1 | ○ | ○ | ○ | ○ | ○ | 01/09/2010 | 4d, 16 h |
| Event 2 | ○ | ○ | ○ | ○ | ○ | 21/08/2009 | 2d, 16 h |
| Event 3 | ○ | ○ | X | ○ | ○ | 11/11/2009 | 5d, 19 h |
| Event 7 | ○ | ○ | X | ○ | ○ | 27/08/2009 | 3d, 16 h |
| Event 8 | ○ | ○ | ○ | ○ | ○ | 28/08/2010 | 1d, 16 h |
| Event 9 | ○ | X | ○ | ○ | ○ | 19/12/2009 | 1d, 16 h |
| Event 13 | ○ | ○ | ○ | ○ | ○ | 26/12/2010 | 2d, 16 h |
| Event 19 | X | ○ | ○ | X | ○ | 26/10/2012 | 5d, 16 hrs |





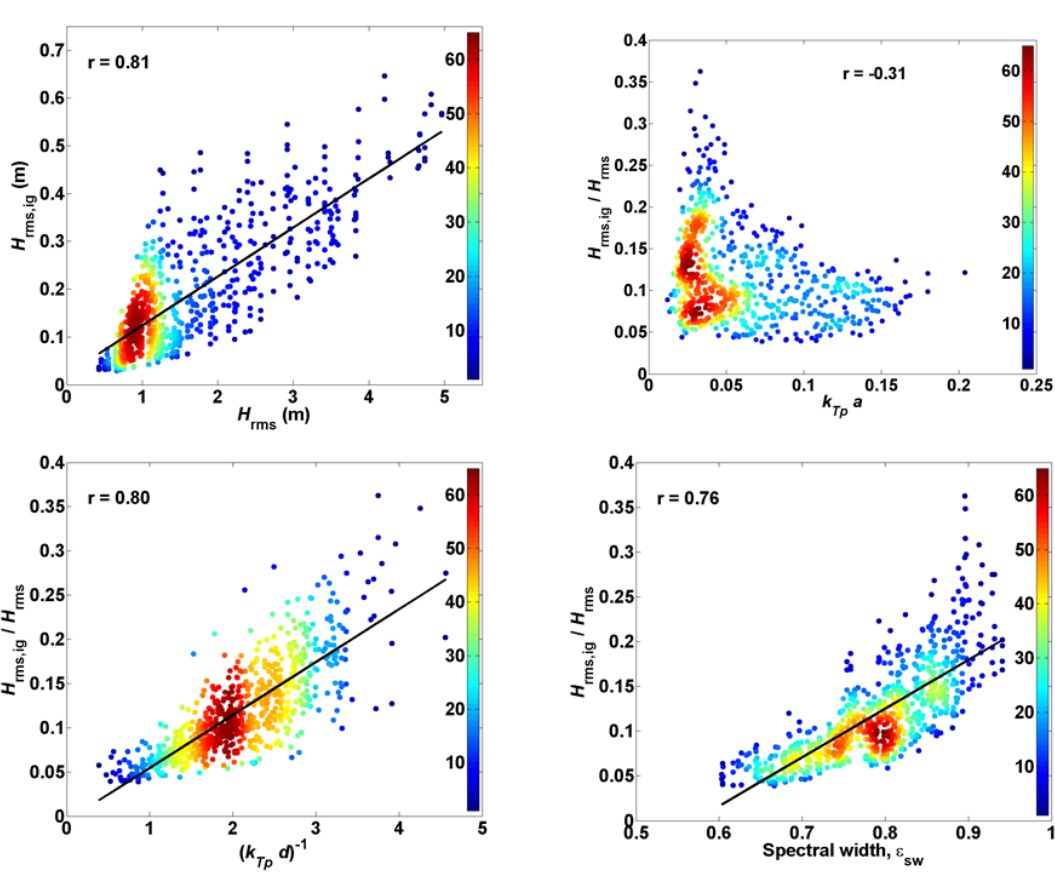

**Figure 4.** Density scatter plots at FRF site, AWAC data: Wave heights (top left), wave steepness (top right), relative depth (bottom left), and spectral width (bottom right).




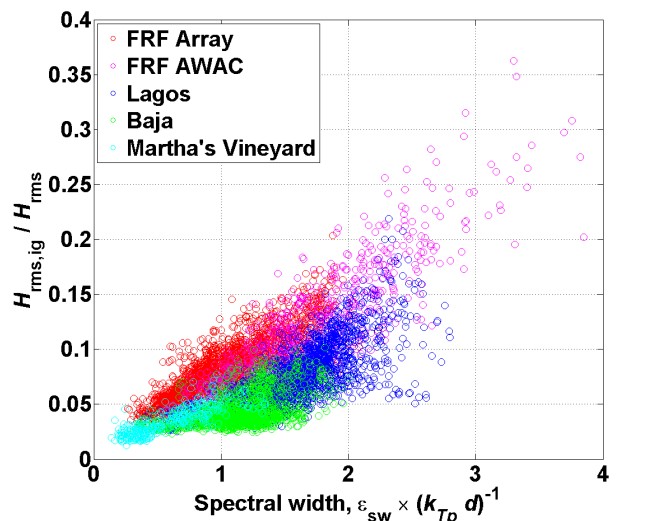

**Figure 5.** Relative depth and spectral width scatter plot for all data.

**Figure 6.** Predicted infragravity wave heights based on regression analysis.

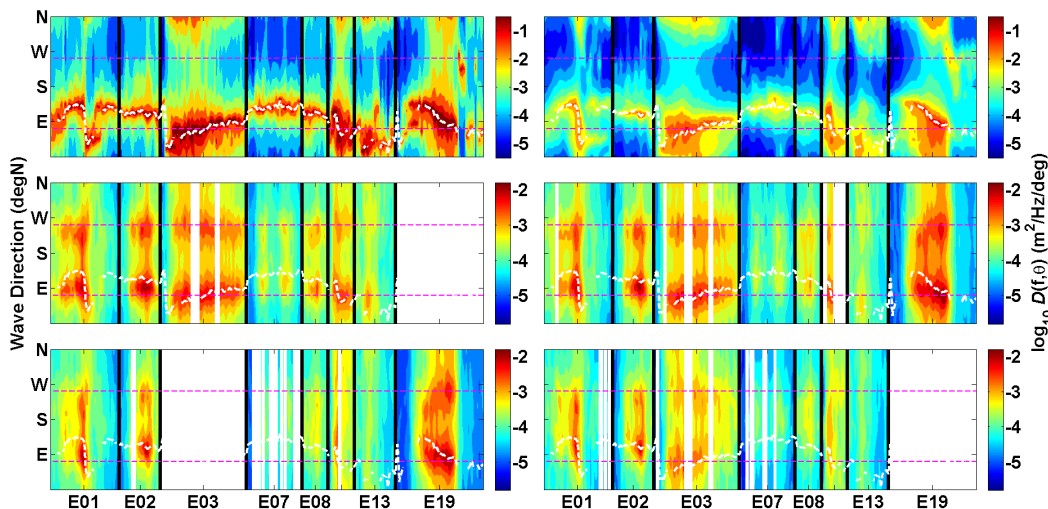

**Figure 7.** Peak (top left) and mean (top right) wind wave, and AWAC5m (middle left), AWAC6m (middle right), AWAC8m (bottom left) and AWAC11m (bottom right) infragravity wave directional spectrograms for FRF AWAC data. The white line depicts the peak wind-generated wave directions, and the dashed magenta lines represent the shore-normal orientation of the data site.




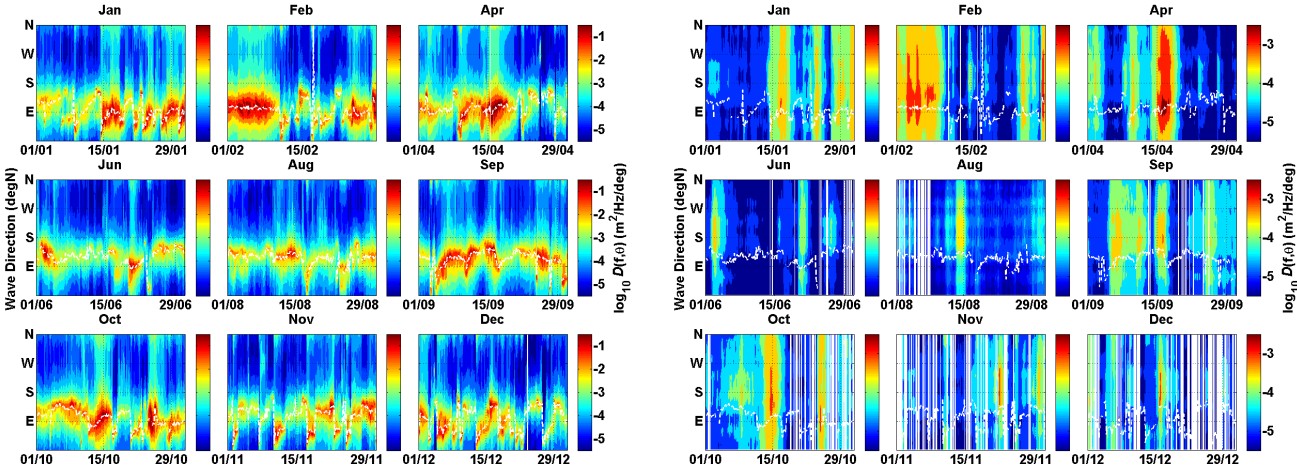

**Figure 8.** Monthly directional spectrograms of wind-generated waves for FRF pressure array data.

**Figure 9.** Monthly directional spectrograms of infragravity waves for FRF pressure array data. The white dashed line depicts the peak wind-generated wave directions.

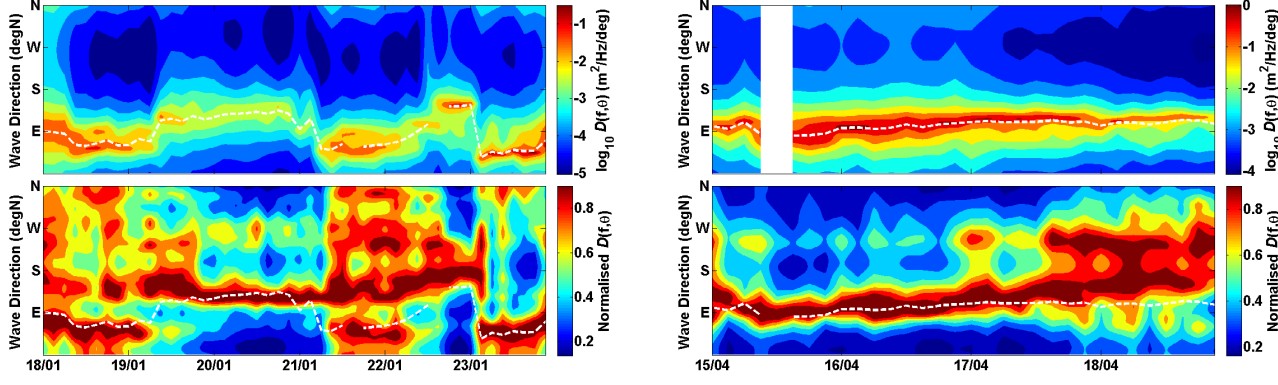

(a) Wind-generated wave (top) and normalised infragravity wave (bottom) directional spectrograms for January event.

(b) Wind-generated wave (top) and normalised infragravity wave (bottom) directional spectrograms for April event.

**Figure 10.** Examples of directional spectrograms for FRF pressure array data.



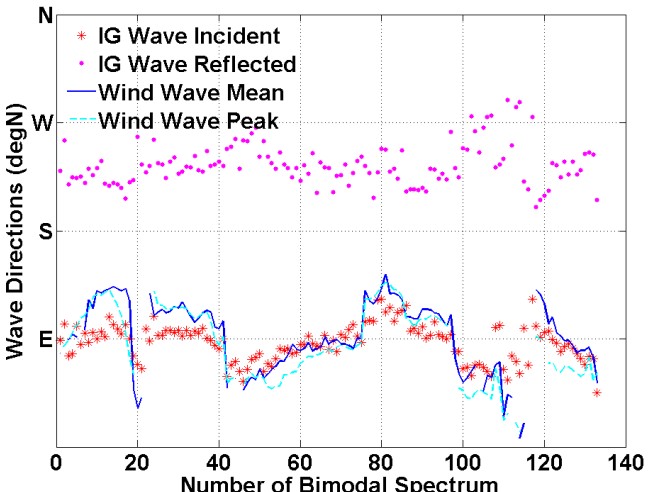

**Figure 11.** Wave directions for FRF AWAC data with bimodal directional spectrum.

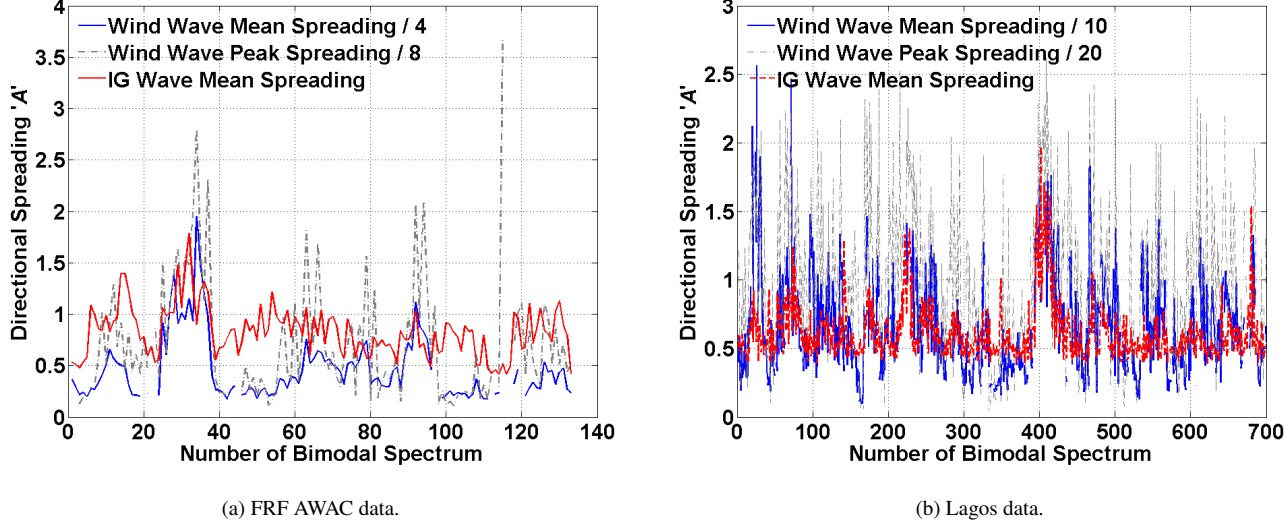

(a) FRF AWAC data.  (b) Lagos data.

**Figure 12.** Directional spreading for wind-generated waves and infragravity waves with bimodal directional spectrum.




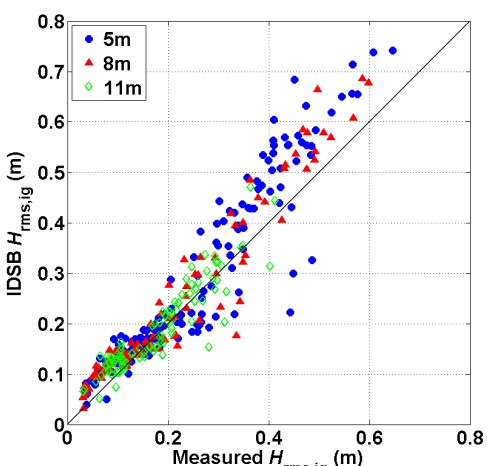

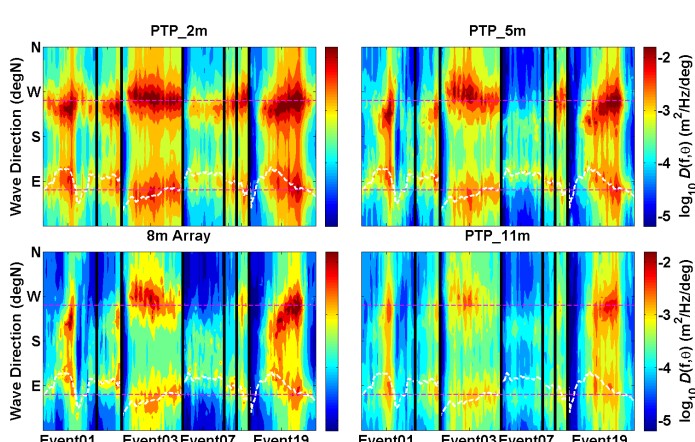

**Figure 13.** Comparison of measured and IDSB modelled infra-gravity wave heights

**Figure 14.** Directional spectrograms for all IDSB modelled data. The white line depicts the peak wind-generated wave directions at the forcing boundary, and the dashed magenta lines represent the shore-normal orientation of the data site.