# Peer review of "In situ observations of infragravity wave directionality at nearshore coastal sites"

_Ocean Science, 2017_

## Referee Comment (RC1) · Anonymous Referee #1 · 9 Dec 2017

General Comments This paper describes the analysis of some nearshore observations of infragravity (IG) waves taken at 4 widely separated locations: southern California, at Martha's Vineyard and the Field Research Facility (FRF) at Duck, on the east coast of USA, and at Lagos in Nigeria. Emphasis is placed on the statement that 'Infragravity wave directional properties outside of the surf zone are seldom studied'. The authors refer to conventional and 'new' directional analysis methods. The objective of the analysis was to obtain improved insight into the IG wave directional properties. This has been done by a correlation analysis of IG wave height and wind wave height, against wave steeness, relative depth and spectral width. It was found qualitatively that when infragravity wave energy increased, their directional distributions become predominantly bimodal, having peaks in both incident (shore-normal) and reflected directions.

Finally the IDSB model of Reniers et al. (2002) was applied, showing some skill in reproducing the observations.

The topic of IG waves is interesting and receiving more attention recently since free IG waves in the deep ocean are thought likely to need taking into account for interpretation of altimeter data e.g. Aucan and Ardhuin (2013). The present paper, however, is rather superficial and poorly written, and needs major revision. It omits many significant references, makes many inaccurate and unsupported statements and the analysis is not very strong, while it also lacks convincing evidence of new results.

Specific Comments 1. Remove the first sentence from the Abstract, it is not helpful or accurate and the definition of IG waves is discussed in the Introduction, it is not part of the original work in this paper. Note that infragravity waves could refer to all gravity waves with longer periods than about 30s, which would include tides, tsunamis, Rossby waves, but is usually taken to refer to those gravity waves between 30-300s period, linked to wind-waves. 2. Section 1: The Introduction needs a through revision. A cursory search of the literature on IG waves reveals guite a number of papers not referred to here. There are some inaccuracies and inconsistencies. For example, contrary to the authors' statement that the physics of free wave generation is not comprehensively understood, Aucan and Ardhuin (2013) state that the liberation of bound IG into free IG waves at the shoreline is now relatively well understood. They refer further to Henderson and Bowen (2002) but they acknowledge it remains a difficult modelling problem (mainly due to difficulty of resolving nearshore length scales). Given their long wavelength, most of the outbound free IG energy is trapped by refraction as edge waves on the shelf, propagating alongshore, and only a small fraction of the IG energy leaks into the open ocean as free waves (Webb et al., 1991). Please carry out a more careful review of the literature, then state the motivation for the present work. What are IG waves important? State what methods are used and what is the aim of the paper. Clarify what is original in the analysis and results. 3. Section 2: Why were

OSD
the specific datasets chosen? Most of the analysis (as acknowledged) focusses on the FRF data. While the Lagos data were referred to, the Martha's Vineyard and Baja data are not presented. What dates were selected (not shown in Table 1)? 4. Section 3 is about frequency analysis and section 4 about directional analysis: There is a lot of detail about fairly standard wave spectral analysis but no clarification as to what is the 'new' analysis method. The methods are not clear and the results are presented under the same section. It would be better to separate the sections. Why is no attempt made to separate free and forced IG waves (these are clearly separated in wavenumber space)? Is there evidence of edge waves other than the statements about complex multi-modal seas? 5. The numerical modelling is wrapped into section 4 where it might be better to have a separate section. The results of the model show the same disparity between model and observations as already identified in Reniers et al. (2010). Technical corrections Please take care with the English, in many places there are superfluous or missing words and unnecessary repetition.

---

## Author Comment (AC1) · 30 Dec 2017

**Response to RC1**

Dear Anonymous Referee #1,

Thank you for reviewing this manuscript. Regarding the general comments, it is true that infragravity (IG) waves are increasingly gaining attention in deep ocean research. Notwithstanding, IG waves are formed and most dynamic in the coastal region, so we believe field data analysis of this area is of interest to the oceanographic community. Furthermore, IG waves in the nearshore has been suggested to play an important role in sediment transport, e.g., formation of longshore bar and beach cusps (Baldock and Huntley, 2002), and are of interest to the industry for loading and offloading of carriers, port development, etc. The below text details our response to the specific comments, and attempts to demonstrate that statements in the manuscript are not inaccurate and unsupported, and outcomes of the analyses are relevant.

- 1. We have removed the sentence in the revised version.
- 2. We understand and acknowledge there are a number of studies regarding the liberation of bound IG waves, e.g., Henderson and Bowen (2002) and Janssen et al. (2003), and the process is reasonably understood through theoretical and experimental studies. Therefore, we should have tempered the language, and this has been revised. However, we would like to show that the physics of this process is still a very complicated issue.

A thorough and comprehensive review of IG wave generation is provided in Baldock (2012), which is referenced in the original manuscript. While noting that the progressive release of bound waves from wind-wave group has been clearly observed in some experiments, e.g., Janssen et al. (2003), he debates evidence of the widely adopted assumption that bound IG waves are released during windwave breaking where reference is often made to Longuet-Higgins and Stewart (1962). Through a review of existing experiments and papers, he describes the implications of IG wave dissipation to the release of bound waves and also reveals that Longuet-Higgins and Stewart (1962) themselves state that bound waves could be reflected before wind-wave breaking, which implies that the bound wave release is not related to the onset of wind-wave breaking. He confirms that there are two different modes of free wave generation, and the dominant mode of generation is understood to be dependent on relative beach slope and wave steepness. We agree that horizontal resolution to simulate IG wave generation is a problem for oceanic scale spectral models (like WAVEWATCH III®) but not so for phase-resolving models. Currently, we still lack an established numerical IG wave model that can accurately simulate IG wave processes in any wave and geographical conditions, although, we note that the first application of the sophisticated SWASH model to field scales with irregular bathymetry is reported in Rijnsdorp et al. (2015). Furthermore, estimates of bound waves become unreliable for large waves as the assumption of a slowly varying wave group is no longer valid (Reniers et al., 2002, 2010). Even sophisticated models such as Herbers and Burton (1997) and Henderson et al. (2006) do not explain large wave events with strong nonlinearity.

Regarding why IG waves are important; Webb et al. (1991) did find that only a small fraction of IG energy leaks into the deep ocean, but Aucan and Ardhuin (2013) state that little is known about typical energy levels and variability of free IG waves in the open ocean. They further point out the IG wave relevance to the open ocean citing references therein, and their investigation led to the development of an IG module in WAVEWATCH III® (Ardhuin et al., 2014). Since IG waves are generated and most dynamic in the nearshore, we believe IG wave field observations in this region is important. We have included the above literature review in more detail, clearly stated the IG wave relevance, and provide with clarity the original and new methods, analysis, and results in the revision. The revision has also been amended to clearly state that there are two analysis components: the frequency analysis and directional analysis components.

3. The manuscript structure has been revised and include a section to provide clear details regarding the selection of data sets and description. The Martha's Vineyard and Baja data are included in the frequency spectrum analysis, but the results were not strong in the directional analysis. This has been clarified in the revision.

- 4. The manuscript structure has been amended to make these points clear for readers. Regarding separating the free and forced IG waves, we refer to Point 1 that the estimates become unreliable for extreme waves, and hence, we focus on total wave energy. We acknowledge that wavenumber spectra could be used to detect edge waves, and that may provide insights into the complex directionality; this has been explored in the revision.
- 5. The revised paper has a separate modelling section. The modelling was undertaken to validate the observations as there is no previous work to show that IG wave directions can be derived from the instruments used in this study. The same disparity between observations and the model raises an interesting point whether the current understanding and model formulation are accurately reproducing reflected wave magnitudes. This point has been clarified and discussed in the revision.

Lastly on the technical comments, the revised manuscript attempts to take care of the superfluous use of English and missing words, which will be submitted in due course. Thank you kindly for your review.

Kind regards,

Takehiko Nose on behalf of all co-authors

**References**

- Fabrice Ardhuin, Arshad Rawat, and Jerome Aucan. A numerical model for free infragravity waves: definition and validation at regional and global scales. *Ocean Modelling*, 77:20–32, 2014.
- Jerome Aucan and Fabrice Ardhuin. Infragravity waves in the deep ocean: An upward revision. *Geophysical Research Letters*, 40(13):3435–3439, 2013.
- TE Baldock. Dissipation of incident forced long waves in the surf zone—implications for the concept of "bound" wave release at short wave breaking. *Coastal Engineering*, 60:276–285, 2012.
- TE Baldock and DA Huntley. Long-wave forcing by the breaking of random gravity waves on a beach. In *Proceedings of the Royal Society of London A: Mathematical, Physical and Engineering Sciences*, volume 458, pages 2177–2201. The Royal Society, 2002.
- Stephen M Henderson and AJ Bowen. Observations of surf beat forcing and dissipation. Journal of Geophysical Research: Oceans, 107(C11), 2002.
- Stephen M Henderson, RT Guza, Steve Elgar, THC Herbers, and AJ Bowen. Nonlinear generation and loss of infragravity wave energy. *Journal of Geophysical Research: Oceans (1978–2012)*, 111(C12), 2006.
- THC Herbers and MC Burton. Nonlinear shoaling of directionally spread waves on a beach. Journal of Geophysical Research: Oceans (1978–2012), 102(C9):21101–21114, 1997.
- TT Janssen, JA Battjes, and AR Van Dongeren. Long waves induced by short-wave groups over a sloping bottom. *Journal of Geophysical Research: Oceans*, 108(C8), 2003.
- Michael S Longuet-Higgins and RW Stewart. Radiation stress and mass transport in gravity waves, with application to 'surf beats'. *Journal of Fluid Mechanics*, 13(04):481–504, 1962.
- AJHM Reniers, AR Van Dongeren, JA Battjes, and EB Thornton. Linear modeling of infragravity waves during delilah. *Journal of Geophysical Research: Oceans*, 107(C10):1–1, 2002.
- AJHM Reniers, MJ Groenewegen, KC Ewans, S Masterton, GS Stelling, and J Meek. Estimation of infragravity waves at intermediate water depth. *Coastal Engineering*, 57(1):52–61, 2010.
- Dirk P Rijnsdorp, Gerben Ruessink, and Marcel Zijlema. Infragravity-wave dynamics in a barred coastal region, a numerical study. *Journal of Geophysical Research: Oceans*, 120(6):4068–4089, 2015.
- Spahr C Webb, Xin Zhang, and Wayne Crawford. Infragravity waves in the deep ocean. Journal of Geophysical Research: Oceans, 96(C2):2723–2736, 1991.

---

## Referee Comment (RC2) · Anonymous Referee #2 · 1 Jan 2018

General comments: This paper claims to provide some new analysis of directional IG waves and reports on some preliminary numerical modelling results. In general the paper lacks focus, misses important relevant references and is very fuzzy it is presentations so that it is impossible to understand the figures and the data that is analyzed, and the general feeling is that the paper contains nothing new compared to Hebers et al. (1994, 1995). Based on the following 3 major concerns I suggest that the paper is rejected and the authors take the time to sort out their ideas and clarify them rather than wasting the time of reviewers.

1) Missing references: there are quite a few recent IG array analysis performed by groups in the UK and the US: Harmon et al. (GRL 2012), Neale et al. (JGR 2015), Godin et al. (JGR 2014) that looked at the directional distribution of IG wave energy,

and Rawat et al. (2014) performed the first trans-oceaning tracking of IG wave events.

2) It is very frustrating that the there are no details on the regression analysis mentioned several times by the authors. What parameters are used? What is the correlation coefficient? If it is Pearson's r, then r=0.93 ... is typically less than reported by Ardhuin et al. (2014). In general page 8 is particularly disappointing. The authors discuss correlation with mean periods Tm02... but why use Tm02 when it is well known that it is the peak which matters? Why not use Tm0,-1 or even the more exotic Tm0,-2 of Ardhuin et al. (2014)? That previous paper provided already a parametric relationship for both the total IG energy and its frequency distribution ... why not use that relation? Is the one proposed here any better?

3. The directional analysis result is not clear al all. For example, on figure 10: which method is used? IMLE usally performs well on large arrays, why not use that one for the FRF (and by the way the directional spectra are already computed by C. Long or K. Hathaway at the FRF...

4. Why chose a non-conventional "A" spreading parameter that cannot be easily measured direcly, unlike Kuik et al. (JPO 1988) sigma which is simply given from a1 and b1 ? this "A" is fairly sensitive to reflections...

Other minor details:

Page 2 line 27: "more skewed toward unity" -> "closer to unity"

Page 3: other datasets could be cited : e.g. Harmon et al. (GRL 2012)

Page 8, line 13: why use Tm02 instead of Tm0,-1 or other ??

Line 19: "we aimed to develop a simple parametric relationship": which one?

Page 10, line 30: Is it the MEM of Lygre and Korgstad (1986) or another estimator?

Page 11: Eq. 14 is not necessary ... and variables a1, b1 are not defined.

[Figure]

Page 19: line 22: "sufficiently resolved" by what?